# Effect of Ti on the Structure and Mechanical Properties of Ti*_x_*Zr_2.5-*x*_Ta Alloys

**DOI:** 10.3390/e23121632

**Published:** 2021-12-04

**Authors:** Bin Zhang, Yu Tang, Shun Li, Yicong Ye, Li’an Zhu, Zhouran Zhang, Xiyue Liu, Zhen Wang, Shuxin Bai

**Affiliations:** Department of Materials Science and Engineering, College of Aerospace Science and Engineering, National University of Defense Technology, Changsha 410073, China; Bin_Zhang1997@163.com (B.Z.); linudt@163.com (S.L.); mr_zla@163.com (L.Z.); materialsryan@outlook.com (Z.Z.); Liuxy85722@163.com (X.L.); wangzhen@nudt.edu.cn (Z.W.); shuxinbai@hotmail.com (S.B.)

**Keywords:** Ti-Zr-Ta alloys, lattice structure, morphology, mechanical properties, mixing entropy

## Abstract

To determine the effects of Ti and mixing entropy (Δ*S*_mix_) on the structure and mechanical proper-ties of Zr-Ta alloys and then find a new potential energetic structural material with good me-chanical properties and more reactive elements, Ti_*x*_Zr_2.5−*x*_Ta (*x* = 0, 0.5, 1.0, 1.5, 2.0) alloys were investigated. The XRD experimental results showed that the phase transformation of Ti_*x*_Zr_2.5−*x*_Ta nonequal-ratio ternary alloys depended not on the value of Δ*S*_mix_, but on the amount of Ti atoms. With the addition of Ti, the content of the HCP phase decreased gradually. SEM analyses revealed that dendrite morphology and component segregation increasingly developed and then weakened gradually. When *x* increases to 2.0, Ti_*x*_Zr_2.5−*x*_Ta with the best mechanical properties can be ob-tained. The yield strength, compressive strength and fracture strain of Ti_2.0_Zr_0.5_Ta reached 883 MPa, 1568 MPa and 34.58%, respectively. The dependence of the phase transformation and me-chanical properties confirms that improving the properties of Zr-Ta alloys by doping Ti is feasible.

## 1. Introduction

Energetic structural materials (ESMs) are a class of multifunctional reactive mixtures combining outstanding strength and energetic characteristics, including high density, high energy density and low sensitivity [1]. ESMs can be initiated upon impact loading and then release a large quantity of chemical energy through self-reaction or reaction with air [2]. Highly efficient damage can be obtained through the combination of mechanical damage derived from the conversion of kinetic energy and extra damage from the release of chemical energy.

In the past decade, ESMs have attracted increasing attention [3,4,5,6]. As an active metal element with lower reaction threshold, Zr is widely used in energetic structural materials [7,8,9]. In addition, in order to improve the penetration ability of the warhead, Ta with high density and certain oxidation energy release is also widely used [10,11]. Therefore, if Zr and Ta can be used together in energetic structural materials, the lethality of ESMs would be greatly improved. However, the mechanical properties of Zr-Ta binary alloy are generally poor due to the existence of HCP phase, which severely restricts its application. Therefore, improving the mechanical properties of Zr-Ta alloys may have significance to ESMs.

According to Ti-Zr and Ti-Ta binary phase diagrams (Figure 1a,b), Ti atoms can completely dissolve in Zr and Ta lattices as single BCC solid solutions in a broad temperature range. Meanwhile, the introduction of Ti can lower the phase transition temperature of Zr and then make β-Zr (BCC structure) more stable [12]. Moreover, according to entropy engineering proposed by Lu et al. [13], alloys with higher mixing entropy tend to form simple solid solutions and show better mechanical properties. Besides, the addition of the most active element Ti can further enhance the lethal power of Zr-Ta alloys.

However, the effects of Ti and mixing entropy on the structure and mechanical properties of Ti-Zr-Ta alloys are controversial. On the one hand, the formation of a BCC solid solution matrix does not mean better mechanical properties. For example, the addition of 5 at% Ti decreased the elastic modulus of a Zr-30%Ta alloy from 99.5 ± 7.2 GPa to 73.6 ± 6.3 GPa [14]. On the other hand, some works found that the high-entropy effect on ternary nonequal ratio alloys should be very weak. For example, compared with Ti-25Ta binary alloys, Ti-25Ta-15Zr ternary alloys do not exhibit a stronger ability to form single-phase solid solutions [15].

In this work, the structure and mechanical properties of Ti*_x_*Zr_2.5-*x*_Ta (*x* = 0.0, 0.5, 1.0, 1.5, 2.0) alloys were investigated. As the relationships of the composition–structure–morphology were established based on the phase diagram, the effects of Ti and mixing entropy on the structure and mechanical properties of Zr-Ta alloy were revealed, and the feasibility of improving the mechanical properties of Zr-Ta alloys by Ti doping was confirmed.

## 2. Materials and Methods

The as-cast Ti-Zr-Ta alloys were vacuum arc melted from a mixture of raw metals with a purity over 99.9% in a Ti-gettered argon atmosphere. The ingots had to be remelted at least eight times to achieve a homogeneous distribution of constituent elements. During each cycle, the sample was flipped and kept in a liquid state for ~5 min.

The crystal structures of the prepared bulk were characterized by X-ray diffraction (Rigaku Dmax 2500) using Cu-Kα radiation with a scanning rate of 10°/min and region between 10° and 90°. The microstructure was examined by scanning electron microscopy (SEM, TESCAN MIRA3 LMH, Czech Republic) equipped with back-scattered electron (BSE) and energy-dispersive spectroscopy (EDS, X MAX20, Oxford, UK) detectors. Specimens for SEM examination were grinded with SiC papers ranging from 400 to 3000 grit, and polished with 5 μm, 3.5 μm, and 1 μm adamantine gypsum, respectively [16]. All polished samples were ultrasonically cleaned to remove grease and other surface contaminants. The quasi-static mechanical tests were performed using a universal testing machine (Instron-3369, Instron, MA, USA) at an initial strain rate of 10^−3^ s^−1^. Compression tests were performed on cylindrical specimens with dimensions of Φ 5 mm × 10 mm. XRD, SEM and compressive specimens were obtained by electrical discharge wire cutting.

The volume fractions of different phases were obtained from SEM-BSE images using Image-Pro software (version Plus 7.0). The route was as following: first, the image except the part of the scale plate was divided into several regions according to their grayscales; secondly, the numbers of pixels belonging to each region were counted and then these values were normalized as percentages of the total number of the pixels; finally, the target phase region belonging to HCP phase, which had been identified comprehensively by XRD and SEM, was sort out and its percentage of pixels represented the volume fraction of HCP phase in the alloy.

## 3. Results

### 3.1. Composition of Ti_x_Zr_2.5-__x_Ta Alloys

According to the EDX results of Ti*_x_*Zr*_2.5-x_*Ta (*x* = 0, 0.5, 1.0, 1.5, 2.0) alloys, the actual compositions of the as-cast alloys are listed in Table 1. Note that the experimental values listed in Table 1 are all the average of the results from at least five different parts of the ingot. The experimental compositions of all as-cast Ti*_x_*Zr*_2.5-x_*Ta alloys were almost the same as the expected values.

### 3.2. Lattice Structure of Ti_x_Zr_2.5-__x_Ta Alloys

Figure 2a shows the XRD patterns of the Ti*_x_*Zr_2.5*-x*_Ta (*x* = 0, 0.5, 1.0, 1.5, 2.0) alloys. As expected, the as-cast Zr_2.5_Ta alloy is composed of HCP and BCC solid solution phases. The introduction of Ti obviously affects the phase and lattice structure of the as-cast alloys. As shown in Figure 2b, the intensity of the diffraction peaks corresponding to the HCP phase decreases gradually as the Ti content increases. When *x* increased to 2.0, the diffraction peaks of the HCP phase disappeared, and only one set of BCC peaks existed. The XRD results indicate that the presence of Ti represses the formation of the HCP phase in Ti*_x_*Zr_2.5-*x*_Ta alloys.

### 3.3. Morphology of Ti_x_Zr_2.5-__x_Ta Alloys

The SEM-BSE images at different scales as well as corresponding EDS point results and mapping images of Ti*x*Zr_2.5-*x*_Ta (*x* = 0, 0.5, 1.0, 1.5, 2.0) alloys are shown in Figure 3. The morphology and composition distribution of Ti*_x_*Zr_2.5-*x*_Ta alloys are dependent on the Ti content. Note that the composition listed in Figure 3 is the average value of more than 3 areas with the same contrast in SEM-BSE images.

For the Zr_2.5_Ta binary alloy, although a uniform microstructure and element distribution were seen from the lower magnification images, a fine lamellar structure can be observed in the higher-magnification images, as shown in Figure 3a.

In general, the morphology and composition distribution of Ti_0.5_Zr_2.0_Ta are similar to those of the alloy without Ti. The macroscopically uniform morphology is composed of finer structures. As shown in Figure 3b, Ta, with the highest atomic mass, is rich in regions around grain boundaries, shown as white regions in the BSE images. Meanwhile, a fine basketweave structure appears inside the grains. Specifically, many spherical precipitates in which the inner lamellar structure radiates from the center to the periphery are distributed in the grains.

As the Ti content increases to *x* = 1.0 and 1.5, the Ti_1.0_Zr_1.5_Ta and Ti_1.5_Zr_1.0_Ta alloys present distinct dendrite morphologies, as shown in Figure 3c,d. EDS mapping shows that dendrite and interdendrite regions are rich in Ta and TiZr, respectively. Additionally, many fine continuous and granular precipitates were distributed in the Ti1.0Zr1.5Ta alloy matrix and then disappeared in the Ti_1.5_Zr_1.0_Ta alloy.

When *x* increased to 2.0, the as-cast Ti_2.0_Zr_0.5_Ta alloy possessed a typical equiaxed grain morphology, as shown in Figure 3e. Ti and Zr are enriched at grain boundaries, and Ta is enriched inside grains. No obvious fine precipitate can be observed in the high-magnification SEM-BSE images.

According to the BSE image and EDS mapping, Ti*x*Zr_2.5-*x*_Ta (*x* = 0, 0.5, 1.0, 1.5, 2.0) alloys are mainly composed of two regions: one is the Zr-rich area, and the other is the Ta-rich area. As it is known, Zr belongs to HCP phase stable element, while Ta is BCC phase stable element. Therefore, the phase with higher Zr content shown as black region in BSE image is easier to form HCP structure [17]. Based on this, the content of HCP phase was quantified by image recognition technology using Image-Pro software. As shown in Figure 2b, the values obtained by BSE statistics are consistent with the XRD result.

### 3.4. Mechanical Properties of Ti_x_Zr_2.5-x_Ta Alloys

Figure 4 shows the compressive mechanical properties of the Ti*_x_*Zr_2.5-*x*_Ta (*x* = 0, 0.5, 1.0, 1.5, 2.0) alloys. The yield strength (σ_y_), compressive strength (σ_c_) and fracture strain (ε) are given in Table 2. As the Ti content increases, the compressive strength and fracture strain of Ti*_x_*Zr_2.5-*x*_Ta (*x* = 0, 0.5, 1.0, 1.5, 2.0) alloys first decrease from 1410 MPa to 1252 MPa and 25.77% to 4.5%, respectively, and then increase to 1568 MPa and 34.58%, respectively. The yield strength displays the opposite trend. In comparison, the Ti_2.0_Zr_0.5_Ta alloy with a single BCC phase and equiaxed grains shows the best mechanical properties. Its compressive strength and fracture strain reached 1568 MPa and 34.58%, respectively.

## 4. Discussion

### 4.1. Effect of the Mixing Entropy

According to entropy engineering proposed by Lu et al. [13], the lattice structure of multicomponent alloys, including ternary non-equimolar alloys, is controlled by the mixing entropy (Δ*S_mix_*), which is calculated by Equation (1) [18].
(1)$$\Delta S_{mix}=-R\overset{n}{\underset{i=1}{\sum }}c_{i}\ln c_{i}$$
where *c_i_* is the atomic percentage of the *i’*th element and *R* is the gas constant. According to the thermodynamic relation $\Delta G=\Delta H_{mix}-T\Delta S_{mix}\;$ [19], the higher Δ*S_mix_* is, the lower the Gibbs free energy Δ*G_mix_*; thus, the high-mixing entropy effect would result in the formation of a single disordered solid solution. However, in Ti*_x_*Zr_2.5*-x*_Ta (*x* = 0, 0.5, 1.0, 1.5, 2.0) ternary alloys, the content of the secondary phase with the HCP structure does not depend on the value of Δ*S*_mix_, as shown in Table 3 and Figure 1b. Thus, there should be other factors determining the phase of Ti*_x_*Zr_2.5*-x*_Ta (*x* = 0, 0.5, 1.0, 1.5, 2.0) ternary alloys.

### 4.2. Effect of Ti on the Lattice Structure of Ti_x_Zr_2.5__-x_Ta Alloys

As shown in the Ti-Zr and Ti-Ta binary phase diagrams (Figure 1a,b), Ti atoms can completely dissolve in the Zr or Ta lattice as a single BCC solid solution in the temperature range of 800 to 1800 K. In contrast, the temperature range of a single BCC solid solution is only 1800–2200 K for Zr_2.5_Ta, as shown in Figure 1c, where the red line represents the composition of the Zr_2.5_Ta alloy. In other words, the presence of Ti broadens the temperature region of a single BCC solid solution in the Zr-Ta binary phase diagram to a lower temperature. In addition, Ti can also lower the phase transition temperature of β-Zr, which makes β-Zr (BCC crystal) more stable. Therefore, the Ti*_x_*Zr_2.5*-x*_Ta alloys tend to form a single BCC solid solution with increasing Ti content, as shown in the XRD results (Figure 2).

Note that the presence of HCP solid solution in the as-cast Zr2.5Ta alloy means that the equivalent temperature of the solidification conditions in this work should be lower than 1100 K. Thus, a single BCC solid solution does not appear in the as-cast TixZr2.5-xTa alloy, and many complex and fine morphologies result, as shown in Figure 3.

### 4.3. Effect of Ti on the Morphology of Ti_x_Zr_2.5__-x_Ta Alloys

Similar to the lattice structure, the morphology of Ti*_x_*Zr_2.5-*x*_Ta (*x* = 0, 0.5, 1.0, 1.5, 2.0) alloys is also dependent on the Ti content, as shown in Figure 3. It is known that noneutectic alloys composed of different constituents have an obvious difference between melting and solidification temperatures, shown as the presence of a liquid–solid two-phase region in the isomorphous phase diagram. The nonequilibrium solidification of these alloys generally has a dendrite morphology and compositional segregation. When the difference between the melting and solidification temperatures is very small or the liquid-solid two-phase region is very narrow, the compositional segregation of the as-cast multicomponent alloy can be neglected. In the Ti*_x_*Zr_2.5-*x*_Ta system, Ti and Zr have the same crystal structure and allotropic transition temperature and can completely dissolve in each other, as shown by the small mixing enthalpy (Table 4). Therefore, Ti and Zr tend to cluster together during solidification, and component segregation of Ti and Zr in multicomponent alloys is slight [20,21].

Ta plays a different role in the Ti*_x_*Zr_2.5-*x*_Ta system. As shown in Figure 1c, the Zr-Ta binary system has a wide liquid–solid two-phase region when the Zr content is lower than 70 at% and a very small difference between the melting and solidification temperatures when the Zr content is higher than 70 at%. Thus, the uniform microstructure and elemental distribution of Zr_2.5_Ta are very good, as shown in Figure 3a. In contrast, Ti-Ta has a typical isomorphous liquid–solid two-phase region. As the Ti/(Ti + Ta) ratio increases from 0 to 66.6 at%, the difference between the melting and solidification temperatures increases first and then decreases, as shown in Figure 2b. Thus, the introduction of Ti also makes the difference between the melting and solidification temperatures of Ti*_x_*Zr_2.5-*x*_Ta alloys increase first and then decrease. Therefore, the dendrite morphology and component segregation of Ti*_x_*Zr_2.5-*x*_Ta alloys increasingly develop as *x* increases from 0 to 1.5 and then gradually disappear as *x* further increases to 2.0, as shown in Figure 3.

### 4.4. Effect of Ti on the Mechanical Properties of Ti_x_Zr_2.5__-x_Ta Alloys

It is known that dendrite morphology and component segregation at the micron scale hinder the movement of dislocations. The obstruction of dislocation movement generally results in a higher yield strength, which is the critical strength for dislocation movement, as well as a lower fracture strain, which is decided by the dislocation movement ability. Therefore, the yield strength and fracture strain of Ti*_x_*Zr_2.5-*x*_Ta alloys display the same and opposite trends as the development and weakening of dendrite morphology and component segregation. More precisely, as *x* increases from 0 to 1.0, the developed dendrites and enhanced component segregation make the yield strength and fracture strain of Ti*_x_*Zr_2.5-*x*_Ta alloys increase and decrease, respectively. When *x* further increases to 2.0, the decreasing obstruction of dislocation movement from dendrites and component segregation substantially increase the fracture strain of Ti*_x_*Zr_2.5-*x*_Ta alloys and decrease the yield strength.

Unlike the heterostructure at the micron scale, dislocations can pass through the boundaries of some fine morphologies [23,24,25]. The pinning of fine boundaries or precipitates would lead to the strain hardening effect. The finer the heterostructure is, the stronger the strain hardening effect, the higher the ultimate strength and the better the fracture strain. Therefore, Zr_2.5_Ta and Ti_2_Zr_0.5_Ta alloys both show a considerable strain hardening effect, as shown in Figure 3. Comparing them, the disappearance of the HCP solid solution with poor plastic deformation contributes to the Ti_2_Zr_0.5_Ta alloy having the highest ultimate strength and fracture strain.

In summary, the mechanical properties of Ti*_x_*Zr_2.5*-x*_Ta alloys are controlled by Ti content through the lattice structure and morphology.

## 5. Conclusions

In this work, the lattice structure, morphology and mechanical properties of Ti*_x_*Zr_2.5-*x*_Ta (*x* = 0, 0.5, 1.0, 1.5, 2.0) alloys were investigated, and the following conclusions were obtained:The phase transformation of Ti*_x_*Zr_2.5-*x*_Ta (*x* = 0, 0.5, 1.0, 1.5, 2.0) nonequal-ratio ternary alloys depends not on the value of Δ*S*_mix_, but on the amount of Ti atoms. The Zr_2.5_Ta alloy is characterized by a BCC + HCP dual phase structure. With the introduction of Ti, the content of the HCP phase decreases gradually.The degree of component segregation becomes increasingly developed as *x* increases from 0 to 1.5 and then gradually disappears as *x* further increases to 2.0, which is closely related to the relative content of Ti-Ta. As the Ti/(Ti + Ta) ratio increases from 0 to 66.6 at%, the dendrite morphology and component segregation of Ti*_x_*Zr_2.5-*x*_Ta alloys first increasingly develop and then gradually disappear.With the addition of Ti, the yield strength first decreases and then increases. However, the trends of compressive strength and fracture strain are completely opposite. When *x* increases to 2.0, the Ti_2.0_Zr_0.5_Ta alloy with a single BCC phase and equiaxed grains shows the best mechanical properties, and its yield strength, compressive strength and fracture strain reach 883 MPa, 1568 MPa and 34.58%, respectively.

## Figures and Tables

**Figure 1 entropy-23-01632-f001:**
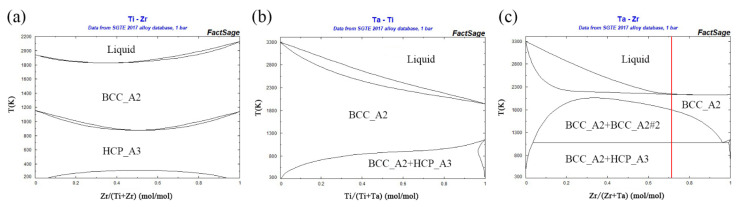
Binary phase diagrams of Ti-Zr (**a**), Ti-Ta (**b**) and Ta-Zr (**c**), the red line represents the composition of the Zr_2.5_Ta alloy.

**Figure 2 entropy-23-01632-f002:**
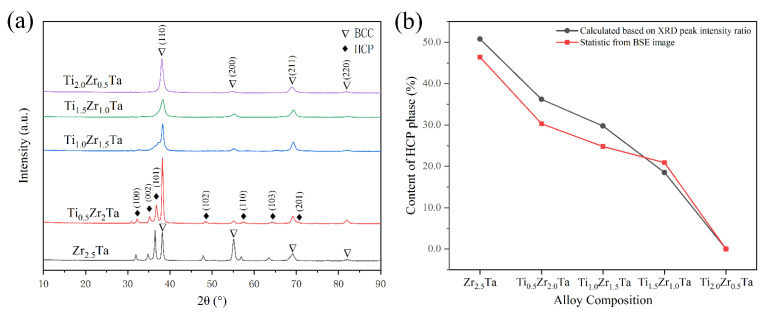
(**a**) X-ray diffraction patterns of the Ti*_x_*Zr_2.5-*x*_Ta (*x* = 0, 0.5, 1.0, 1.5, 2.0) alloys. (**b**) The content of HCP phase calculated based on XRD peak intensity ratio and statistic from BSE image of Ti*_x_*Zr_2.5-*x*_Ta (*x* = 0, 0.5, 1.0, 1.5, 2.0) alloys.

**Figure 3 entropy-23-01632-f003:**
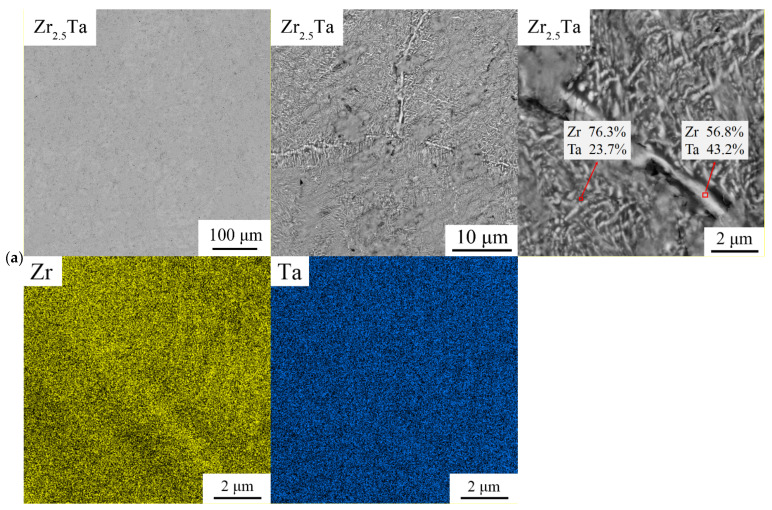
SEM-BSE images and EDS mapping images of Ti*_x_*Zr_2.5*-x*_Ta (*x* = (**a**) 0, (**b**) 0.5, (**c**) 1.0, (**d**) 1.5, (**e**) 2.0) alloys at different magnifications.

**Figure 4 entropy-23-01632-f004:**
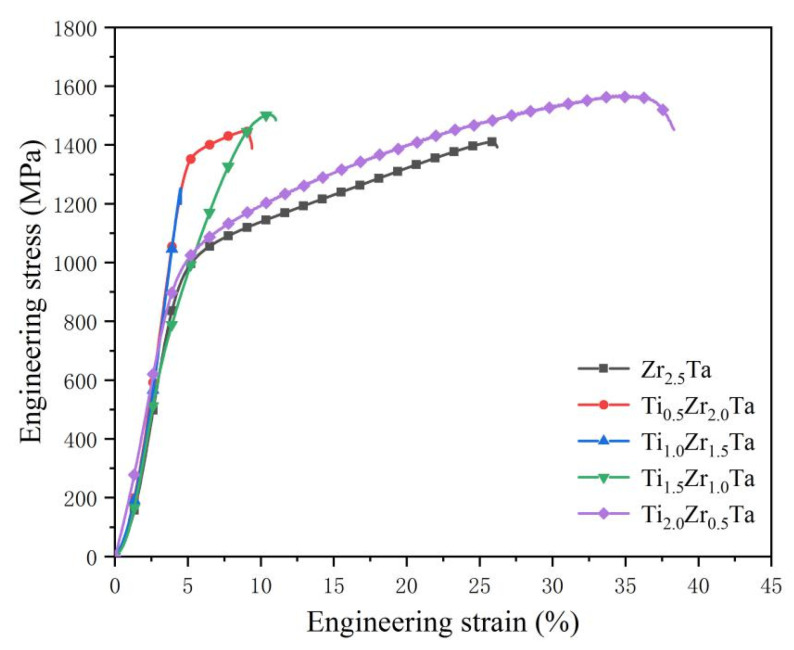
Compressive mechanical properties of Ti*_x_*Zr_2.5-*x*_Ta (*x* = 0, 0.5, 1.0, 1.5, 2.0) alloys at room temperature.

**Table 1 entropy-23-01632-t001:** Designed and actual compositions of the Ti*_x_*Zr_2.5-*x*_Ta (*x* = 0.0, 0.5, 1.0, 1.5, 2.0) alloys.

Alloy		Ti (at%)	Zr (at%)	Ta (at%)
Zr_2.5_Ta	Designed	\	71.4	28.6
	Actual	\	72.8 ± 0.2	27.2 ± 0.2
Ti_0.5_Zr_2.0_Ta	Designed	14.3	57.1	28.6
	Actual	13.8 ± 0.1	59.3 ± 0.1	26.9 ± 0.2
Ti_1.0_Zr_1.5_Ta	Designed	28.6	42.8	28.6
	Actual	28.0 ± 0.2	45.3 ± 0.6	26.7 ± 0.5
Ti_1.5_Zr_1.0_Ta	Designed	42.8	28.6	28.6
	Actual	41.9 ± 0.1	30.4 ± 0.2	27.7 ± 0.1
Ti_2.0_Zr_0.5_Ta	Designed	57.1	14.3	28.6
	Actual	59.9 ± 0.4	12.1 ± 0.4	28.0 ± 0.8

**Table 2 entropy-23-01632-t002:** Yield strength (σ_y_), compressive strength (σ_c_) and fracture strain (ε) of Ti*_x_*Zr_2.5-*x*_Ta (*x* = 0, 0.5, 1.0, 1.5, 2.0) alloys.

Alloy	σ_y_ (MPa)	σ_c_ (MPa)	ε (%)
Zr_2.5_Ta	886	1410	25.77
Ti_0.5_Zr_2.0_Ta	1298	1447	8.86
Ti_1.0_Zr_1.5_Ta	1252	1252	4.50
Ti_1.5_Zr_1.0_Ta	735	1502	10.58
Ti_2.0_Zr_0.5_Ta	883	1568	34.58

**Table 3 entropy-23-01632-t003:** Mixing entropy and phase structure of Ti*_x_*Zr_2.5*-x*_Ta (*x* = 0, 0.5, 1.0, 1.5, 2.0) alloys.

Alloy	Zr_2.5_Ta	Ti_0.5_Zr_2_Ta	Ti_1.0_Zr_1.5_Ta	Ti_1.5_Zr_1.0_Ta	Ti_2_Zr_0.5_Ta
ΔS_mix_	0.598R	0.956R	1.079R	1.079R	0.956R
Phase structure	BCC + HCP	BCC + HCP	BCC + HCP	BCC + HCP	BCC

**Table 4 entropy-23-01632-t004:** Mixing enthalpy (ΔH_mix_) of Ti, Zr and Ta [22].

Element	Ti-Zr	Ti-Ta	Zr-Ta
**ΔH_mix_ (kJ/mol)**	0	1	3

## Data Availability

The datasets used or analyzed during the current study are available from the corresponding author on reasonable request.

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
