# Peer review of "Effect of Ti on the Structure and Mechanical Properties of TixZr2.5-xTa Alloys"

_entropy, 2021, doi:10.3390/e23121632_

Round 1
Reviewer 1 Report
This manuscript could be considered for publication after essential revision.
English should be revised carefully.
Page 2: improve the quality/resolution of fig. 1.
Page 2, lines 55-56. Insert more details.
Page 3: improve the quality/resolution of fig. 2.
Page 5, line 81: Could you specify for SEM micrographs: magnification, acceleration voltage, working distance, and image resolution?
Page 5, line 82: If possible, could you specify the other statistical parameters of 3-D surface roughness, according to with ISO 25178-2: 2012 that are defined as follows:
- a) Height parameters are a class of surface finish parameters that quantify the Z-axis perpendicular to the surface.
(Sq) – root mean square height is the standard deviation of the height distribution, or RMS surface roughness.
(Ssk) – Skewness is the third statistical moment, qualifying the symmetry of the height distribution.
Negative skew indicates a predominance of valleys, while positive skew is seen on surfaces with peaks.
(Sku) – Kurtosis is the fourth statistical moment, qualifying the flatness of the height distribution.
For spiky surfaces, Sku > 3; for bumpy surfaces, Sku < 3; perfectly random surfaces have kurtosis of 3.
(Sp) - Maximum peak height is the height between the highest peak and the mean plane.
(Sv) - Maximum pit height is the depth between the mean plane and the deepest valley.
(Sz) - Maximum height is the height between the highest peak and the deepest valley.
(Sa) - Arithmetical mean height is the mean surface roughness.
Kindly insert a paragraph with Statistical analyses, and explain the method, the software used, and all the parameters related to these statistical experiments.
Page 6, line 106: insert the corresponding reference for “..............proposed by Lu et al.,....”
Page 6, line 112: specify with a range of values for “....is slight.”
Kindly insert references for all mathematical formulas.
Authors may consider citing the following reference:
[1] Ş. Ţălu, Micro and nanoscale characterization of three-dimensional surfaces. Basics and applications. Napoca Star Publishing House, Cluj-Napoca, Romania, 2015.
Reviewer 2 Report
The manuscript focuses on the addition of Ti to the Zr-Ta alloys to find a new potential energetic structural materials with good mechanical properties. Idea of the work is straightforward and clear, however, the manuscript is raw in some sections and should be improved, formally and scientifically. The manuscript is well organized, however, a major revision is mandatory before considered to be published. The main comments are:
- English should be improved in the whole manuscript. A native speaker should go over the text.
- The „Introduction“ section is not coherent and should be rewritten and enriched (e.g. the definition of ESM is not sufficient; other reactive elements besides Ta and Zr suitable for ESM; the more detailed description including possible applications would be welcomed; phases alfa, beta and omega should be characterized in more details).
- Lines 21-22: “In the Zr55Ni5Al10Cu30 system, the content of reactive elements is only 55 at%; thus, the energetic release characteristics of Zr55Ni5Al10Cu30 bulk metallic glasses are unsatisfactory.” What is the expected limit that can be considered satisfactory?
- Line 39: „On the other hand, the mechanical properties of Zr-Ta binary alloys with all reactive elements are generally poor due to the existence of brittle intermetallics and HCP solid solutions.“
- In the particular phase diagram, no intermetallic phases, just solid solutions can be found. Please, explain.
- Could be the mechanical properties of the binary Zr-Ta alloys improved by a heat treatment (e.g. annealing in the BCC region and subsequent quenching)? Since you are looking for alternatives with a high content of reactive elements and good mechanical properties, this question takes place.
- Green rectangle or red vertical line highlighted in Fig. 1 is discussed in the “Discussion” section. Both shapes should be mentioned in the figure caption as well.
- The information about detailed experimental procedure is missing and should be added (e.g. metallographic procedure of the sample preparation for SEM – grinding, polishing, etching; sample form for XRD – bulk or powder?)
- Standard deviations are required, since average values are given in Table 1. Due to the accuracy of EDS method, values should be given to 1 decimal place. Check the values in individual rows (e.g. the sum of the values in the row is higher than 100% for the Ti1.5Zr1.0Ta alloy).
- The bracket with values x is not necessary in the title of chapters 3.1 – 3.4.
- Miller indices of particular phases should be included in XRD patterns.
- The quality of SEM images is not sufficient and should be improved. The images should also comprise the phase assignment resulting from XRD. The addition of particular alloy denotation to the image groups would be also welcomed.
- The EDS maps does not show relevant information, since the magnification is too low for this analysis. The images should be replaced by the maps taken at higher magnification. This will gain the information about the distribution of elements in the microstructure. What was the acquisition time of the mapping?
- In the Ti2.0Zr0.5Ta sample, XRD confirmed the presence of BCC only, however, in the corresponding SEM image (e.g. Fig. 3e with 2μm scale mark), two alternating constituents (dark and bright) within the grains seem to appear. How would you explain this?
- How many samples were tested for the each alloy to avoid anomalies resulting for instance from internal defects, microstructure heterogeneity? What is your opinion related to the values of compressive strength and fracture strain in the case of the Ti1.0Zr1.5Ta alloy? I recommend repeating the test.
- In my opinion, the approach with entropy of mixing is incorrect / insufficient. You consider the ideal mixing only and the interactions between particular elements in the alloy are neglected. These interactions expressed by interaction parameters in thermodynamic databases affect the value of entropy and thus also the value Gibbs free energy. You should reflect this fact in the manuscript. Please, clarify, how you obtained the values of mixing enthalpy of Ti, Zr and Ta.
- The discussion about the addition of Ti to Zr-Ta is theoretical, based on the comparison of the equilibrium phase diagrams. However, the investigated as-cast samples are at non-equilibrium conditions. Why the authors simply did not perform the DTA/DSC analysis? It would bring the evidence on the real temperatures of phase transitions. Such results would enrich the discussion and also improve the scientific level of the manuscript.
Reviewer 3 Report
This manuscript deals with an experimental investigation of TixZr2.5-xTa (x=0, 0.5, 1.0, 1.5, 2.0) alloys. New knowledge concerning phase constitutions and mechanical properties of investigated alloys has been obtained. The manuscript is acceptable for publication after the following minor revisions:
20-21: You mentioned that „Zr and Ta are widely used in ESMs”, but there is no reference where Ta is used in ESMs. Please add some references.
48 Replace the word “must” with “had to”.
63: Table 1: Designed composition of Zr in Ti1.5Zr1.0Ta alloy should be 28.57 and not 42.86. Add standard deviations for the actual composition of all alloys to better see how good the homogeneity of the alloys was.
71: Part b) in Fig. 2 is confusing. Make only one y-axis and show the peak intensity ratio of both phases on only one scale for all alloys.
72: Replace the word “strength” with “intensity” in the figure caption.
78: Fig. 3: I think the SEM- BSE images in Fig. 3 do not match the XRD results. Can you explain it? Please, add a phase assignment obtained by XRD to the images to better see the correspondence with XRD experiments.
112-113: I think the content of the secondary phase with the HCP structure doesn’t depend on the value of ΔSmix at all, not only slightly. Change the sentence.
122: Change “transition temperature of Zr” to “transition temperature of β-Zr”
128: Why "Unlike"? Not rather "Similar to"?
Round 2
Reviewer 1 Report
Accept in present form.
Author Response
Thank you very much.
Reviewer 2 Report
Authors significantly improved the quality of article. However, the major revision is still mandatory. There are two main reasons:
- The microstructural analysis is incomplete. In my opinion, the combination of results obtained from SEM/EDS and XRD with binary or ternary phase diagrams should be sufficient enough to assign the identified phases to particular regions, at least in the case of majority phases (BCC and HCP). The assignment of small precipitates would be probably difficult without SAED. Since you observed alloys in the as-cast conditions, some concentration gradient can be expected within particular regions. However, the distinct phase / structural components regions can be observed in SEM images. To assign BCC and HCP phases to the particular constituents, I would recommend the following procedure:
- to perform the point EDS analysis of individual phase regions (where possible),
- to estimate the volume fraction of phases in the SEM images and compare results with the quantitative XRD analysis,
- to combine the results with the information taken from phase diagrams. Since isothermal sections of Ti-Zr-Ta phase diagram are not available, the Ti-Zr-Nb phase diagram could be roughly used as both Zr-Ta and Zr-Nb binaries are the same in principle. Despite of the as-cast condition of observed samples, this approach could help in the assignment.
- The EDS mapping does not reflect the phase distribution in the observed area. For example, the Zr2.5Ta alloy consists of BCC and HCP phases based on XRD data. In the corresponding SEM-BSE image, two microstructural constituents can be distinguished. However, the EDS maps taken at low magnification show almost homogeneous distribution of Ti, Zr and Ta. In this form, therefore, the maps do not provide any relevant information about the distribution of elements in particular phases as it could be seen at higher magnification maps. Another fact: The microstructure was observed in the regime of back-scattered electrons based on atomic contrast. However, the particular elemental map (e.g. Fig. 3c) does not reflect the chemical composition observed in the SEM-BSE image where dendrites have different chemical composition compared to interdendritic regions. Therefore the elemental maps at higher magnification should be included. These maps could also help in the microstructural analysis described in the previous comment. Maps in Figs. 3d and 3e could be considered acceptable, although the longer acquisition time would increase their quality. For mapping I recommend at least 20 minutes per map with deadtime about 20-30% and acquisition rate around 2/3 of the scale (relevant for the INCA software).
Besides the above comments I would also recommend following minor revisions:
- to round the vaules of chemical composition (designed as well as those obtained from EDS) including their standard deviations to 1 decimal place;
- to distinguish grinding and polishing in the section of the specimens preparation for the SEM examination.
Round 3
Reviewer 2 Report
According to the authors' answers to comments and recommendations, the resubmitted manuscript can be accepted in the present form.